# Psychometric properties of the Post Traumatic Stress Disorder Checklist for DSM-5 (PCL-5) in Greek women after cesarean section

**Eirini Orovou**[1]*, **Irina Mrvoljak Theodoropoulou**[2], **Evangelia Antoniou**[1]

**1** Department of Midwifery, University of West Attica, Aigaleo, Greece, **2** Department of Psychology, National and Kapodistrian University of Athens, Aigaleo, Greece

☙ These authors contributed equally to this work.
* eorovou@uniwa.gr

**Data Availability Statement:** All relevant data is included within the manuscript and Supporting Information files.

**Funding:** We have no funders.

## Abstract

The aim of this study was to examine psychometric properties of the revised Posttraumatic Stress Checklist (PCL-5) for Diagnostic and Statistical Manual– 5th Edition (DSM-5) in Greek postpartum women after Cesarean Section(CS) (emergency-elective).So far, there was no study in Greece assessing psychometric properties of the PCL-5 in women after CS. The participating women ($N = 469$), who gave birth with emergency and elective CS at the Greek University Hospital of Larisa, have consented to participate in two phases of the survey and completed self-report questionnaires, the 2nd day after CS and at the 6th week after CS. Measures used in this study were the PCL-5 for DSM-5, the Life Events Checklist (LEC-5), Criteria B, C, D, E, and Criterion A, specifically designed for detection of posttraumatic stress disorder (PTSD) symptoms in postpartum period. To evaluate the internal reliability of the PCL-5 two different indices of internal consistency were calculated, i.e., Cronbach's alpha (.97) and Guttman'ssplit-half (.95), demonstrating high reliability level. The data were positively skewed, suggesting that the reported levels of PTSD among our participants were low. Factor analyses demonstrated acceptable construct validity; a comparison of thePCL-5 with the other measures of the same concept showed a good convergent validity of the scale. Overall, all the results suggest that the four-factor PCL-5 seemed to work adequately for the Greek sample of women after CS.

## Introduction

The Posttraumatic Stress Disorder (PTSD) is a mental health problem that some people develop after experiencing or witnessing a life-threatening event like combat, disaster, assault or sexual violence (Criterion A) [1]. The most common PTSD symptoms are clustered into a four-dimensional structure: Intrusion/re-experiencing (Criterion B), avoidance (Criterion C), negative alterations in cognition and mood (Criterion D) and alteration in arousal and reactivity (Criterion E) [2–4].

One of the most frequently used PTSD measure is the Posttraumatic Stress Disorder Checklist (PCL) which has recently undergone substantial revision in the 5th Edition of Diagnostic

**Competing interests:** The authors have declared that no competing interests exist.

and Statistical Manual (DSM-5), developed by Weathers et al, 2013 [5]. The PTSD was formally recognized in the Diagnostic and Statistical Manual 3$^{rd}$ version (DSM-III) in 1980 [6], but its diagnostic criteria have been repeatedly revised in the editions of the DSM. Several important revisions were made to the PCL in updating it for the DSM-5.Thus, according to the DSM-5 [2], PTSD has been moved out of the anxiety disorders section and moved to a new category identified as "trauma and stressor-related disorders" [7].The continuing evolution of the field of symptomatology highlights that PTSD remains a complex disorder making accurate measurement of symptoms even more important [8]. Furthermore, a new dissociative subtype of PTSD was created. This subtype, in addition to meeting full PTSD criteria, captures persons who meet dissociative symptoms (depersonalization or derealization) and emotional detachment [9].

The prevalence of PTSD is two times greater in women than in men; it is influenced by hormonal disorders, stressful life events, such as sexual abuse [10,11] and childbirth experiences [12]. For several years a childbirth experience was viewed by scientists as a positive experience for women. Nevertheless, in recent year several studies have shown that childbirth can become a traumatic event; about 1.5%-6% of mothers may develop Postpartum PTSD (P-PTSD) [13,14]. Many researchers also have identified the kind of delivery, and in particular cesarean delivery, as a major risk factor for the development of P-PTSD [15–17], with a great correlation of PTSD to emergency cesarean section (EMCS) and elective cesarean section (ELCS) [18,19]. Other main risk factors for P-PTSD area history of previous mental disorder [19,20], preterm labor [19–22], inclusion of neonate in neonatal intensive care unit [23,24], lack of breastfeeding [19,25,26] and a lack of support from spouse during the perinatal period [14,19,23].The factors that can lead to P-PTSD are common in postpartum women of all countries. Given that CS is associated with P-PTSD [27–29], in Greek mothers the problem is greater due to the high prevalence of CS. Actually, every year in Greece, more than half of deliveries are cesarean deliveries [30] deteriorating the problem.

Furthermore, women who undergo CS are more likely to develop postpartum mental disorders [31,32] and especially P-PTSD compared to women who undergo vaginal birth [16,33] and, therefore, deserve a closer follow-up during the postpartum period [34,35].This great difference between vaginal delivery and CS, given the increasing rate of CS in Greece and worldwide, created a need for a validated measure of P-PTSD after CS. Nowadays, there is a lack of valid scale that measures P-PTSD after CS. Previously, research about P-PTSD has typically adapted questionnaires for use in groups such as militaries and veterans, which may not be applicable to postpartum women after CS. Research in the postpartum period, comparing general PTSD measures with specific postpartum PTSD measures, shows that an agreement between these measures on the identification of diagnostic cases of P-PTSD is low [36]. Considering the increasing rate of postpartum women after a cesarean delivery, a properly adjusted questionnaire is necessary to investigate this special population.

The purpose of this study was to examine the psychometric properties of the PCL-5 in postpartum women after CS in Greece. Specifically, the aim was to assess the consistency and reliability of the PCL-5, the construct validity trough factor analyses, and convergent validity using correlations between the PCL-5 and other measures of PTSD, in a sample of women after CS. For several reasons, this task was far more challenging than the translation of a psychometric instrument for different populations, because the established standards for such procedures cannot be transferred to this population. So far there are no studies assessing the psychometric properties of the PCL-5, in its Greek translation, applied on a special group of postpartum women. However, we expect a good match of the four-factor PCL-5, being applied to the current study sample.

## Material and methods

### Participants

Participants were all postpartum women who underwent a CS (N = 469), with a mean age 32.58 ± 6.15(SD) years. A proportion of them with an ELCS was 61.4% (N = 288), while 38.6% (N = 181) had an EMCS. The majority of them were married (88.5%, N = 415), while 10.0% (N = 47) were engaged or in relationship, and 1.5% (N = 7) were single or divorced. Overall, 43.4% (N = 204) of the sample completed undergraduate or/and postgraduate studies, with similar number of them who finished a high school (N = 196, 41.8%); 10.7% (N = 69) participants finished primary school or junior high. Most of them were from an urban area (79.5%, N = 373) and Greeks (90.2%, N = 423). The financial status for more than half of the participants was middle (68.2%, N = 320), low for 29.0% (N = 136) of them and high for 2.8% (N = 13) of the sample. All postpartum mothers gave their written consent for their participation in the survey and had medical charts from which the past and current medical data were obtained. Excluded from the survey were underage mothers, women who had difficulties in understanding the Greek language or other difficulties at a cognitive level which could create a problem in understanding the psychometric tools. Additionally, women who used psychotropic substances or drugs were excluded from the study to meet the PTSD Criterion E, according to the DSM-5 rules.

### Procedure

The data was collected in two stages, the 2nd day after CS and at the 6th week after CS. During the first stage, after informing the women about the purpose of the study and the importance of their participation, socio-demographic and medical data were collected, along with possible past-traumatic life events and birthtrauma. All women were informed on how to complete the questionnaires, and afterwards they completed them without the presence of others in their room. During the second stage, which was arranged for the end of the postpartum period, the PCL-5 was administrated via a telephone interview, again, without the presence of other people.

### Ethics

The study was approved by the University Hospital of Larisa Ethics Commission. Approval: 18838/08-05-2019.

### Measures

All measures were translated into the Greek version by two Greek-English bilingual researchers, reviewed by an accredited translator, and back translated by an English-Greek bilingual researcher, all without prior knowledge of the research project. Therefore, it involved 2-part translation process—an initial translation by two bilingual researchers, followed by a full review of that translation by a translator, in order to confirm completeness, pick up typos or inadvertent translator error, check the accuracy of the translation and review quality of expression. Afterward, it was back translated to English by another bilingual researcher, to confirm that is accurate.

**Socio-demographic questionnaire.** The research-made screening form includes items on social, demographic, obstetric, neonatal, mental characteristics and previous medical history of the participants. It also included information about the experience of a traumatic CS.

**Life Events Checklist-5 (LEC-5) of DSM-5 [37].** The Life Events Checklist (LEC-5) is a self-report measure designed to screen past traumatic life events in a person's life. The LEC-5

is the most widely used self-report measure for adults, composed of 17 items; each item represents traumas, such as natural disasters, accidents, assaults, sexual violence or other stressful life events. It is the only screening tool that responders can determine different levels of exposure to a traumatic event using five nominal levels of answers:"happened to me", "witnessed it", "learned about it", "part of my job", "not sure" and "does not apply" [37–40].

**Criterion A of DSM-5 [3].** Criterion A, the first of the 8 previously mentioned criteria (A, B, C, D, E, F, G& H) that must be met for the PTSD diagnosis, is met when a person was exposed to death, threatened death, serious injury or sexual abuse in one the following ways: (a) direct exposure, (b) witness to the event, (c) information of the event, and (d) exposure in the working area [1,41]. However, for the purpose of this research, criterion A has been adapted with appropriate questions that determine the exposure of the mother or infant to death, threatened death, or serious health complications of both, according to the requirements of the DSM-5.

**Post-traumatic stress checklist (PCL-5) of DSM-5 [7].** The PCL-5 is a20-item self-report psychometric tool, which was developed to measure and evaluate PTSD symptoms. The scale includes 20 items, rated on a 5-point Likert type scale. The responses are categorized as1 (not at all), 2 (a little bit), 3 (moderately), 4 (quite a bit) and 5(extremely), with regards to traumatic life experiences [7,18]. The scale is composed out of four factors: intrusion/re-experiencing, avoidance, negative alterations in cognition and mood, and arousal and reactivity. A provisional PTSD diagnosis can be made if a score of each item is 2 or above (range 0 to 4), when there is a score of one or more in the categories of criteria B and C and two or more in categories D and E, and by summing the score (range 0–80) for each of the 20 items [7,9,18]. Cronbach's alpha for the PCL-5 in other studies ranged from .76 to .97 [42–45]; alpha coefficient reached an index of .97 in the present study.

## Data analyses

The 20-item PCL-5 version was first psychometrically examined for possible extreme skewness, multicollinearity, large measurement errors and poor reliability potential. All analyses were conducted using the statistical data processing packages SPSS 22.0 and AMOS 22.0.There were no missing data. Following that, the reliability indices were measured, and after wards the construct validity trough factor analyses, and divergent validity, by calculating the correlations between the PCL-5 total and factor scores, and scores from the other measures used in the current study. Exploratory factor analyses (EFA) were conducted to examine the factors' loadings, while a series of confirmatory factor analyses (CFA) models were conducted in order to determine the latent structure that best fit PTSD symptoms as measured by the PCL-5.

## Results

Descriptive statistics and reliability indices of the scale are presented in Table 1. To evaluate the internal reliability of the PCL-5, two different indices of internal consistency were calculated (i.e., Cronbach's alpha &Guttman's split-half), and the mean inter-item correlations to estimate item-to-scale homogeneity.

A standard of values greater than .70 [46] was set for both forms of internal consistency coefficients. As shown in Table 1, Cronbach's α coefficients for the total PCL-5, along with four latent factors, were very good (from .79 to .97). Guttman's split-half coefficients were similar, additionally showing great internal consistency of the scale (from .79 to .95), relating the present sample. Furthermore, inter-item correlations, ranging from .414 to .763, showed very good item-to-scale homogeneity. The correlations between factors were also strong, ranging

**Table 1. Descriptive statistics and reliability coefficients for the PCL-5.**

| Scales (N = 469) | M | SD | Alpha | Split-half | N |
|---|---|---|---|---|---|
| Intrusion/Re-experiencing | 2.71 | 4.52 | .93 | .88 | 5 |
| Avoidance | 1.05 | 1.82 | .79 | .79 | 2 |
| Negative Alterations in Cognition and Mood | 3.33 | 5.38 | .91 | .83 | 7 |
| Alterations in Arousal and Reactivity | 2.29 | 4.20 | .90 | .88 | 6 |
| Total PCL Score | 9.39 | 14.93 | .97 | .95 | 20 |

between .794 and .943. Reported levels of PTSD among participants were low, since the data were positively skewed.

## Factors validity

Initially, it was decided to follow a quick-screening of the PCL-5on exploratory factor analysis basis, in order to determine whether the same factors as the theoretically expected ones would be observed. This analysis studied whether all the items are sufficiently reliable, and whether the four factors were observed in this sample, showing good construct validity. Factor loadings are shown in Table 2.

**Table 2. Exploratory factor analysis for the 20 PCL-5 items.**

| | Items (N = 469) | Factors | | | |
|---|---|---|---|---|---|
| | | 1 | 2 | 3 | 4 |
| 1. | Repeated, disturbing, and unwanted memories of the stressful experience? | **.751** | .285 | .207 | .308 |
| 2. | Repeated, disturbing dreams of the stressful experience? | **.584** | .319 | .287 | .291 |
| 3. | Suddenly feeling or acting as if the stressful experience were actually happening again (as if you were actually back there reliving it)? | **.763** | .334 | .259 | .211 |
| 4. | Feeling very upset when something reminded you of the stressful experience? | **.772** | .270 | .286 | .289 |
| 5. | Having strong physical reactions when something reminded you of the stressful experience (for example, heart pounding, trouble breathing, sweating)? | **.724** | .307 | .383 | .153 |
| 6. | Avoiding memories, thoughts, or feelings related to the stressful experience? | **.446** | .436 | .427 | .335 |
| 7. | Avoiding external reminders of the stressful experience (for example, people, places, conversations, activities, objects, or situations)? | **.609** | .419 | .324 | .167 |
| 8. | Trouble remembering important parts of the stressful experience? | **.477** | .327 | .416 | .201 |
| 9. | Having strong negative beliefs about yourself, other people, or the world (for example, having thoughts such as: I am bad, there is something seriously wrong with me, no one can be trusted, the world is completely dangerous)? | .230 | .381 | .245 | **.762** |
| 10. | Blaming yourself or someone else for the stressful experience or what happened after it? | .333 | .207 | .325 | **.786** |
| 11. | Having strong negative feelings such as fear, horror, anger, guilt, or shame? | .513 | .178 | .369 | **.533** |
| 12. | Loss of interest in activities that you used to enjoy? | .427 | **.629** | .229 | .340 |
| 13. | Feeling distant or cut off from other people? | .284 | **.804** | .250 | .242 |
| 14. | Trouble experiencing positive feelings (for example, being unable to feel happiness or have loving feelings for people close to you)? | .279 | **.817** | .237 | .177 |
| 15. | Irritable behavior, angry outbursts, or acting aggressively? | .458 | **.590** | .330 | .200 |
| 16. | Taking too many risks or doing things that could cause you harm? | .205 | .238 | **.618** | .224 |
| 17. | Being "superalert" or watchful or on guard? | .458 | .425 | **.560** | .194 |
| 18. | Feeling jumpy or easily startled? | **.534** | .188 | .504 | .360 |
| 19. | Having difficulty concentrating? | .379 | .287 | **.696** | .223 |
| 20. | Trouble falling or staying asleep? | .253 | .182 | **.802** | .228 |

The factors extracted are those that had given values greater than 1, a criterion of primary importance for determining the number of factors (Kaiser-Guttman criterion). The analysis was performed by the method of principal component analysis (PCA), with varimax axis rotation and Kaiser normalization. Commonly used PCA method combines manifest (observed) variables into weighted linear combinations that end up as components, where the component correlations and component scores match exactly. It seeks to create optimized weighted linear combinations of variable. Varimax rotation is most often used method as well, as it maximizes the variance within a factor in a way that greater loadings are increased and smaller are minimized [47]. In order to control the presence of some weak items (with unsatisfactory loadings), an exploratory factor analysis was initially performed with the principal axis factoring method in order to provide starting values. No items were excluded.

According to the Kaiser Meyer Olkin criterion (KMO = .96) for the PCL-5 the sample was suitable for further analysis, as well as the table of interrelationships of the 20 questions, according to the Bartlett criterion ($\chi^2$ = 8265.30, $df$ = 190, $p < .001$). As theoretically expected, four factors emerged to which 74.12% of the variance is attributed. Factor loadings, though, appeared to function somewhat differently than the theoretically expected factor structure. The first factor, intrusion, originally having 5 items, explains 25.63% of the total variance and, relating to this study sample, consists of 9 items (all 5 items of intrusion, 2 items of avoidance, 1 item of negative alterations in cognitions and mood, and 1 item of alterations in arousal and reactivity).The second factor, negative alterations in cognitions and mood I, consists of 4 items (3 items of negative alterations in cognitions and mood factor, and 1 item of alterations in arousal and reactivity), differing from the originally expected 7-item loadings, and explains 18.00% of the total variance. The third factor, alterations in arousal and reactivity consist of 4 items, instead of 6 as in prototype scale, explains 17.70% of the total variance, while the fourth factor, negative alterations in cognitions and mood II, explains 12.79% of the total variance of the PCL-5 and consists of 3 items, instead of 7, as previously mentioned. However, it seems that all items load satisfactory, but to some extent were differently recognized from this research sample, with the factor avoidance, consisting of 2 items, not being recognized at all, and the factor negative alterations in cognitions and mood being split in two. In addition, it is observed that item 6 - "Avoiding memories, thoughts, or feelings related to the stressful experience?", loads in the first three factors with a higher value by .01 and .02 in the first factor. Item 7 - "Avoiding external reminders of the stressful experience (for example, people, places, conversations, activities, objects, or situations)?", loads in the first two factors with a higher value by .19 in the first. Next, item 11 –"Having strong negative feelings such as fear, horror, anger, guilt, or shame?", has a higher value by .04 in the fifth factor, comparing to the first, while item 12 –"Loss of interest in activities that you used to enjoy?", has a higher value by .20 in the second, comparing to the first factor. Subsequently, item 15 - "Irritable behavior, angry outbursts, or acting aggressively?" has a higher value by .13 in the second, comparing to the first factor. Finally, item 17 "Being "superalert" or watchful or on guard?" loads in the first three factors with a higher value by .10 and .14 in the third, while item 18 "Feeling jumpy or easily startled?" has a higher value by .04 in the first, comparing to the third factor. However, after removing the items that load in two/three factors at the same time, it is observed that the percentage of variance explained does not change significantly (the difference is 2.04%), therefore, the factor's structure remained as presented.

In the next phase, confirmatory factor analyses were employed for the prototype scale [48], and the following criteria and indices were evaluated: $\chi^2$ (chi-square), $\chi^2/df$ (chi-square over degrees of freedom), RMSEA (root mean square error of approximation), RMR (root mean square residual), GFI (goodness of fit index), CFI (comparative fix index), and TLI (Tucker-Lewis index -in comparison with the null model) and AIC (Akaike information criterion -in

comparison with the null model) (Table 3). For GFI, CFI, and TLI, typically values over .90 indicate adequate model fit, whereas values over .95 indicate good model fit [49].

When the same set of data is used for an exploratory and for a confirmatory factor analysis, a sense of cyclical research practice is created. However, in a confirmatory analysis, correlations (loadings) of items within a factor to which they belong are maximally controlled, while loadings of factors to which items do not belong are considered as zero value and are controlled based on this matter. Also, from a substantive point of view, exploratory and confirmatory factor analysis should lead to the same conclusions when applied to the same data. However, models are different, as in an exploratory factor analysis (or in principal component analysis) non-zero correlations of items with all dimensions that arise in data are allowed. If, therefore, confirmatory factor analysis cannot confirm results of exploratory factor analysis on the same data (which is often the case), one cannot expect confirmatory factor analysis to confirm results of exploratoryfactor analysis in a different sample or population [50]. For these reasons, we present here both the exploratory and the confirmatory analyses results, with special emphasis certainly on the confirmatory approach.

It is observed that the original four-factor structural model, $\chi^2(164) = 1014.64$, $p < .001$, RMSEA = .105 (confidence intervals .099 - .112), RMR = .038, GFI = .82, CFI = .90, TLI = .88, AIC = 8439.92 was not fully confirmed by our data (Table 3).However, after model modification, with 45 error covariance, which differ from zero and strictly withinfactors, and which we identified through the modification indices, the model of the PCL-5 with 20 questions, $\chi^2(126)$ = 434.79, $p < .001$, RMSEA = .60 (confidence intervals .058 - .75), RMR = .024, GFI = .92, CFI = .96, TLI = .94, AIC = 602.79, was satisfactory confirmed by our data. For best-fitting model, we decided to allow for those error covariances to differ, as confirmatory factor analysis can include error covariances designating that two measures covary due to other than the shared factor's influence, such as method effects [51]. As expected, due to the large sample size, chi-square remained statistically significant. Still, RMSEA, RMR, AIC and $\chi^2/df$ decreased, whereas GFI, TLI and CFI increased. We could have dropped some items to enhance factor consistency and reduce error levels, thus stabilizing the solution, but this would have negative effects on the factors' theoretically expected validity, leading to a theoretically non-defensible

**Table 3. Confirmatory factor analysis of the PCL-5.**

|  | *M1* | *M2* | *M3* |
|---|---|---|---|
| $\chi^2$ | 8399.92 | 1014.64 | 434.79 |
| *df* | 190 | 164 | 126 |
| *p* | $p < .001$ | $p < .001$ | $p < .001$ |
| $\chi^2/df$ | 44.21 | 6.19 | 3.49 |
| RMSEA [90% CI] | .304 [.298, .309] | .105 [.099, .112] | .060 [.058, .075] |
| RMR | .527 | .038 | .024 |
| GFI | .13 | .82 | .92 |
| CFI | .000 | .90 | .96 |
| TLI | .000 | .88 | .94 |
| AIC | 1106.64 | 8439.92 | 602.79 |
| $\Delta x^2$ |  | 7385.28 | 7965.13 |
| $\Delta df$ |  | 26 | 64 |
| *p* |  | $p < .001$ | $p < .001$ |

Note: M1 –independence model, M2 –four-factor model, M3 –modified four-factor model, including error covariance estimates strictly within factors.

solution. However, the PCL-5 conforms to the original dimensions contained in the scale, as proposed by the creators of the scale (Fig 1). In other words, these results suggest that the four-factor structure of the PCL-5 seems to work adequately for the Greek sample of women after c-section.

## Scale validity

Due to the above-mentioned scale skewness, Spearman's correlations were used to assess convergent scale validity. Namely, to evaluate validity of the PCL-5 in the context of this study, the relationship between known predictor variables of PTSD and the PCL-5 was examined. Moreover, there were significant positive correlations between symptoms of PTSD investigated by the PCL-5 scale, with A, B, C, D, E post-traumatic stress disorders criteria and Life Event Checklist (LEC-5).Concretely, to evaluate the PCL-5 associations with the PTSD criterions, we examined correlations between both total and factor PCL-5 scores, and B, C, D, E criterions and LEC-5, indicating presence of PTSD symptoms and known to be strongly associated with exposure to traumatic events, as well as with criterion A, specifically created for this study. As presented in Table 4, all the correlations were strong, ranging from .443 to .835, showing that diverse types of traumatic events were significantly positively correlated with the PCL-5, and implying very good convergent scale validity, relating the sample of the present research.

## Discussion

The aim of this study was to fill an important gap in the literature by evaluating the psychometric properties of the PCL-5 among Greek women after CS. In fact, there is no reliable and valid screening instrument for postpartum PTSD [52,53], therefore this scale was used to determine the presence of a PTSD diagnosis in women after birth via caesarian section. Nevertheless, the PCL-5 is one of the most commonly used self-report measures of PTSD symptoms, but, to our knowledge, this is the first study in Greece using the PCL-5 questionnaire in general as a screening tool for postpartum PTSD.

The Cronbach's alpha and Guttman'ssplit-half coefficient, as well as factors and item-total correlations were computed to assess the internal consistency and homogeneity of the PCL-5. Some aspects of validity were examined too. Construct validity was examined trough exploratory factor analyses, to see whether the four-factor scale specified by the DSM-5 was observed in this sample. To complement the EFA, confirmatory factor analyses were conducted in order to evaluate the fit of four-factor model, identified in the literature. Furthermore, the convergent validity assessed by Spearman's correlation coefficient was evaluated by examining correlations between the PCL-5 total and factor scores and specific measures regarding the same concept.

Overall, our results support a very good factor structure of the PCL-5, relating to the sample of the current study. The analyses which were satisfactory, as assessed with both Cronbach's alpha and Guttman'ssplit-half coefficients and the mean inter-item correlations, indicated that PCL-5 has highly acceptable reliability and homogeneity. Finally, the scale also demonstrated good construct and convergent validity. Concretely, it was reported an excellent Cronbach's alpha value of .97 for the total score, comparing to the values for the PCL-5 of other studies, ranging from .76 to .97 [42–45]. However, the above studies concerned PTSD in other populations and not specifically in postpartum women; i.e. women after CS. Additionally, the current study provided valuable evidence for the construct validity of the PCL-5 with a four-factor structure, though presenting some differentiation of factor loadings in EFA, in relation to the scale as proposed in the theory. It can be presumably referred to some culture perception differences, or lacking of more focused items targeting this specific moment in life of a woman.

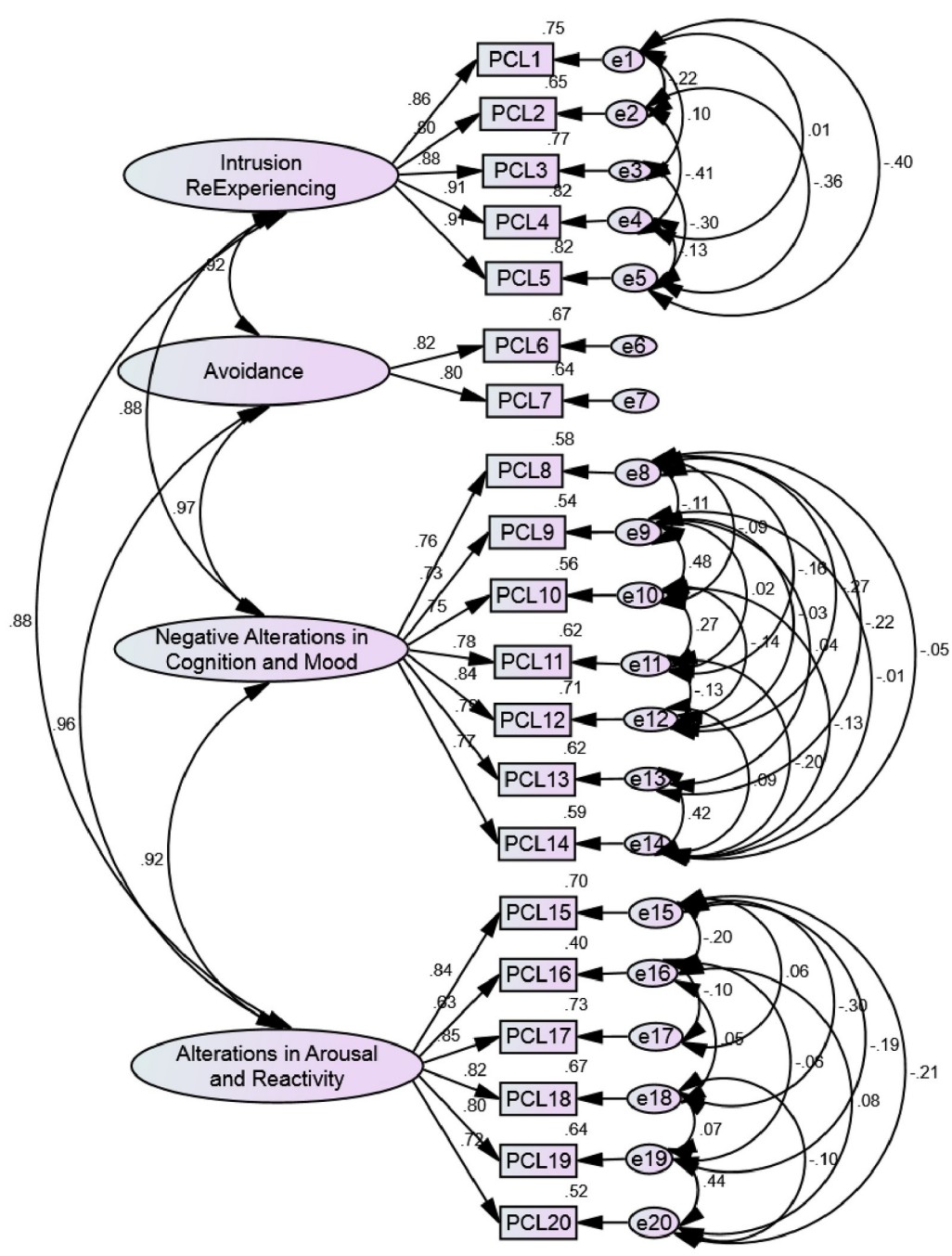

$x^2= 434.79$

$df= 126$

$p<.001$

RMSEA = .060(.058 - .075)

GFI = .92

CFI = .96

**Fig 1. Confirmatory factor analysis outcomes: Graphic representation of the PCL-5 factor model.**

**Table 4. Bivariate Spearman rho correlations among PCL-5 factors and PTSD criterions and LEC-5.**

|  | PCL-5 Total Score | Intrusion | Avoidance | Negative Alterations in Cognitions and Mood | Alterations in Arousal and Reactivity |
|---|---|---|---|---|---|
| Criterion A | .545** | .554** | .518** | .478** | .506** |
| Criterion B | .781** | .789** | .742** | .742** | .656** |
| Criterion C | .812** | .750** | .835** | .772** | .729** |
| Criterion D | .802** | .720** | .704** | .827** | .712** |
| Criterion E | .801** | .728** | .705** | .727** | .828** |
| LEC-5 | .477** | .439** | .443** | .452** | .452** |

**. Correlation is significant at the 0.01 level (2-tailed).

Our findings however, are in line with some results from prior studies, reporting moderate factor structure discrepancy, when comparing to the theoretical scale construct [54,55]. Quite many error covariance sin CFA could be explained by this discrepancy, but good four-factor model fit seems to be evident and applicable to the current study population. The results do not refer to any cut-off score, because in practical use, the optimal cut-off score should be considered cautiously; a woman who was screened positive may require further investigation to confirm PTSD diagnosis.

Using exploratory and confirmatory factor analytic approaches, the multi-dimensionality of the PCL-5was surely confirmed, which is consistent with the majority of studies conducted in primary care settings [54–56], or general population-based research setting [57,58]. The scale also demonstrated convergent validity, with high correlations, ranging from .477 to .812, with A, B, C, D, E post-traumatic stress disorders criterions and the Life Event Checklist.

Generally, reviewing related studies on some other populations, psychometric properties of the PCL-5 appeared also satisfactory, as among female Filipino domestic workers, where high Cronbach's alpha demonstrated excellent scale reliability, with very good convergent validity and high correlations with depression, generalized anxiety, rumination, and direct trauma exposure [59]. The results of the studies on trauma-exposed college students [60] and US veterans [61] indicated good internal consistency, satisfactory reliability, and significant correlations with measures of other constructs (convergent and discriminant validity). Also, the first validation study of the Malay version of a PCL-5 for Malaysian fire and rescue officers demonstrates a valid and reliable scale for screening probable PTSD diagnosis [62]. However, it seems that the results of the current research follow the related studies on other populations who experienced traumatic events.

## Conclusions

This study has several strengths. First it is applied to the population clearly enduring a highly traumatic experience, but lacking a valid screening instrument for measuring PTSD symptoms, especially after an urgent CS. Second, it is a relatively large sample size, and implementation of a demanding analytic plan. Furthermore, the study expands the literature by including assessment of the Greek-language version of the PCL-5 among women in postpartum period, after CS. Such evidence is also important in mental health intervention delivery, particularly in assessing change and intervention effectiveness. The study provides a potential foundation for further investigations into mental health and trauma after CS. Despite these strengths, some limitations must be considered when interpreting the study findings. The most evident limitation is the absence of gold standard measures for validation analyses and to better support the findings of the convergent scale validity. Also, the current study does not provide data for discriminant (divergent) validity of the scale. Additionally, it could be also interesting to add

some postpartum trauma related question/s, in order to analyze the original PCL-5 scale, supported by such item/s. Also, there is a lack of evidence for measuring discriminate validity. Nevertheless, despite those yet unanswered question, the results suggest that the Greek language version of the PCL-5 may be used as a screening tool in postpartum period after CS, as it presented very strong reliability and good structural and convergent validity.

As expected, positively skewed data suggested low reported levels of PTSD among participants, that is common finding for this population [18]. The reason of including both emergency and elective CS groups in the study, although it might appear as a study limitation, was dramatically increased reported maternal morbidity with CS compared with vaginal delivery [63]. Of course, it could be beneficial to analyze psychometric properties of the PCL-5 of only those that undergone emergency CS. However, satisfactory results of the current study could suggest that such analyses could demonstrate even more fitting results. Finlay, further studies that evaluate validation, the utility of correct classification, as well as determination of the optimal cut-off score of, would provide more evidence, relevant for measuring PTSD symptoms in postpartum period.

## Supporting information

**S1 Database.**
(SAV)

## Acknowledgments

The authors would like to thank Mrs. Maroula Paulatou, Dr Dimitrios Xirofotos and Ms. Orestes Papazisis, who helped in the translation of the PCL-5, LEC-5 and Criterion A, and Professor Alexandros Daponte, dean of Medical Faculty of University of Thessaly, for his help to collect the research sample.

## Author Contributions

**Conceptualization:** Eirini Orovou, Irina Mrvoljak Theodoropoulou, Evangelia Antoniou.

**Data curation:** Eirini Orovou, Irina Mrvoljak Theodoropoulou, Evangelia Antoniou.

**Formal analysis:** Irina Mrvoljak Theodoropoulou.

**Investigation:** Eirini Orovou, Evangelia Antoniou.

**Methodology:** Eirini Orovou, Evangelia Antoniou.

**Project administration:** Eirini Orovou.

**Resources:** Eirini Orovou, Evangelia Antoniou.

**Software:** Irina Mrvoljak Theodoropoulou.

**Supervision:** Evangelia Antoniou.

**Validation:** Evangelia Antoniou.

**Visualization:** Eirini Orovou, Irina Mrvoljak Theodoropoulou, Evangelia Antoniou.

**Writing – original draft:** Eirini Orovou.

**Writing – review & editing:** Eirini Orovou, Irina Mrvoljak Theodoropoulou, Evangelia Antoniou.

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
