## [Decision Letter · Decision Letter 0]

13 May 2021

PONE-D-21-05505

Psychometric Properties of the Post Traumatic Stress Disorder Checklist for DSM-5 (PCL-5) in Greek Women after Cesarean Section

PLOS ONE

Dear Dr. Orovou,

Thank you for submitting your manuscript to PLOS ONE. After careful consideration, we feel that it has merit but does not fully meet PLOS ONE’s publication criteria as it currently stands. Therefore, we invite you to submit a revised version of the manuscript that addresses the points raised during the review process.

 I appreciated the importance of your work. My own and reviewer's comments are attached. Please address all points raised here.

We look forward to receiving your revised manuscript.

Kind regards,

Kenta Matsumura

Academic Editor

PLOS ONE

Journal Requirements:

In line with PLOS' guidelines on detailed reporting (https://journals.plos.org/plosone/s/criteria-for-publication#loc-3), please ensure that you have provided sufficient detail on the participants in the Methods section, including the total sample size and relevant demographic characteristics

4. We noticed you have some minor occurrence of overlapping text with the following previous publication(s), which needs to be addressed:

- https://link.springer.com/article/10.1186/s12888-017-1304-4?code=11af2b49-e623-4f24-980b-5d328c5346f8&error=cookies_not_supported

- https://www.tandfonline.com/doi/full/10.1080/20008198.2019.1571378

In your revision ensure you cite all your sources (including your own works), and quote or rephrase any duplicated text outside the methods section. Further consideration is dependent on these concerns being addressed.

Additional Editor Comments:

Fig 1: Coefficients and lines are overlapping and hard to see. Values for PLC5, 14, 20 are missing. Please specify whether the values are standardized parameter estimates or not in the caption.Why didn't the authors use oblique rotation in the explanatory factor analysis? In contrast, why did the authors use principal component analysis? Please refer to Osborne, J. W. Best Practices in Exploratory Factor Analysis, CreateSpace Independent Publishing, Scotts Valley, 2014. I think the authors do not need to reanalyze data but should mention these issues anywhere in the manuscript.The result shows low discriminant validity; i.e., factors should not correlate too high (≥ 0.85 is considered problematic according to Brown, T. A. Confirmatory Factor Analysis for Applied Research 2nd ed., Guilford Press, New York, 2015). Although the authors say, "Also, there is a lack of evidence for measuring discriminate validity." in the conclusion section, please also mention this issue elsewhere in the manuscript.Please check the fair usage of a technical term such as 'discriminate' vs. 'discriminant' validity and 'factors' vs. 'factor' vs. 'factorial' validity.Please check the in-text citation style.

Reviewers' comments:

Reviewer's Responses to Questions

**Comments to the Author**

1. Is the manuscript technically sound, and do the data support the conclusions?

Reviewer #1: Yes

Reviewer #2: Yes

Reviewer #3: Yes

2. Has the statistical analysis been performed appropriately and rigorously? 

Reviewer #1: Yes

Reviewer #2: Yes

Reviewer #3: Yes

3. Have the authors made all data underlying the findings in their manuscript fully available?

Reviewer #1: Yes

Reviewer #2: Yes

Reviewer #3: Yes

4. Is the manuscript presented in an intelligible fashion and written in standard English?

Reviewer #1: Yes

Reviewer #2: Yes

Reviewer #3: Yes

5. Review Comments to the Author

Reviewer #1: The authors interpret the acceptable utility of a PTSD questionnaire among Greek women after Caesarean section. The authors present the high reliability, internal consistency and an appropriate factor structure which is logic. The research design appropriate and the high number of recruited study persons allows them to make stable conclusions. However, there are some minor remarks:

The Introduction section is too long. I think that the detailed description of PTSD touching all criteria are unnecessary and should be focused on those details that are appropriate to the results. The historical presentation of PTSD is absolutly an unnecessary surplus in the MS. PTSD is relatively common entity in perinatal context, but the description of the PTSD symptoms, questionnaires should be more concise.

In the Materials and Methods section:

'Medical dossier' is is not a good expression. Medical charts are the proper word.

In the Ethics sub-section: the second meaning is unnecessary ('After ...')

If the authors carried out test-retest reliability, then they should present the results. Were there any back and forth translation and pilot testing of the questionnaire to test the comprehension?

Description of LEC-5, DSM-5 and PCL-5 can be shortened.

In Data Analyses: 'Descriptive statistics ...' sentence is unnecessary.

The Discussion is a little short and may complete with PCL-5 Factor analysis results in other populations and the consequence of the factor structure.

Reviewer #2: I read it with interest. Examining the need for mental care after cesarean section is expected to become increasingly important in the future, and the results of this study will considered to be useful findings in clinical practice.

I think it need to mention about the limitations of this study more. For example, the author states that "Reported levels of PTSD among participants were low, since the data were positively skewed" in the results, but the effect of this on the results has not been considered. In addition, the participants were two types by Caesarean Sections received, and the impact of this on results should be stated.

Reviewer #3: Thank you very much for valuable opportunity for peer review of this article. The authors conducted examination to psychometric properties of the revised Posttraumatic Stress Checklist (PCL-5) for DSM-5 in Greek postpartum women after Cesarean Section. PTSD in perinatal period is a socially important issue around the world. There is a lack of evidence on the Greek version of PCL-5, so the authors' research questions are important. However, I think it is necessary to reconsider the several concern before publishing it as an academic paper. I look forward to careful and appropriate revisions from authors.

Major concerns:

1)Please describe the translation process of the evaluation scale in more detail. Please show the back translation version of PCL-5 and consider the difference from the original version of one. Moreover, please consider the Linguistically limitations of the Greek version of PCL-5. Also, if there are any points that you paid particular attention to during the translation process, please describe them. In addition, please add the Greek version of PCL-5 as an attachment file (an image file is also acceptable).

2)The strength of this study is the large number of samples. On the other hand, too many samples are statistically more likely to cause type I errors (prone to false statistically significant differences). Please estimate the sample size required for this study from prior probabilities and consider whether the number of samples is appropriate.

Minor concern:

1)Please describe a brief review of the psychosocial characteristics of the Greek postpartum women after Cesarean Section, comparing them to other countries.

6. PLOS authors have the option to publish the peer review history of their article (what does this mean?). If published, this will include your full peer review and any attached files.

Reviewer #1: **Yes: **Zoltan Kozinszky MD, PhD, Sweden

Reviewer #2: No

Reviewer #3: No

---

## [Author Response · Author response to Decision Letter 0]

3 Jun 2021

Replies to Editor comments 

Dear Editor, 

Thank you for your comments on my manuscript, which helped me a lot to revised it. I quote below the corrections that were made. 

• Fig 1: Coefficients and lines are overlapping and hard to see. Values for PLC5, 14, 20 are missing. Please specify whether the values are standardized parameter estimates or not in the caption. 

We tried to open the figure as much as possible in order all the coefficients to be more observable. It is an automatic configuration and it is not possible to put every coefficient on its line and there is an impression that some estimations were missing, although they were not. We do hope that the appearance is satisfying now. Thank you for the chance to improve it. 

• Why didn't the authors use oblique rotation in the explanatory factor analysis? In contrast, why did the authors use principal component analysis? Please refer to Osborne, J. W. Best Practices in Exploratory Factor Analysis, CreateSpace Independent Publishing, Scotts Valley, 2014. I think the authors do not need to reanalyze data but should mention these issues anywhere in the manuscript.

We used PCA with varimax rotation as in Salleh, M. N. M., Ismail, H.,&Yusoff. M. H.(2020). Reliability and validity of a posttraumatic checklist-5 (PCL-5) among fire and rescue officers in Selangor, Malaysia. Journal of Health Research. When using oblique rotation, we receive both a pattern matrix and structurematrix. This seems to be a source of much confusion in practice. Also, some authors have argued that oblique rotations produce less replicable resultsas they might overfit the data to a greater extent. We added an explanation just above Table 2. Exploratory Factor Analysis for the 20 PCL-5 items, with your suggested reference. Thank you for the opportunity to add such explanation. 

• The result shows low discriminant validity; i.e., factors should not correlate too high (≥ 0.85 is considered problematic according to Brown, T. A. Confirmatory Factor Analysis for Applied Research 2nd ed., Guilford Press, New York, 2015). Although the authors say, "Also, there is a lack of evidence for measuring discriminate validity." in the conclusion section, please also mention this issue elsewhere in the manuscript.

This sentence was wrong. We corrected it: ‘’Also, the current study does not provide data for discriminant (divergent) validity of the scale.’’Thank you very much. 

Correlations in Table 4 represent convergent validity, not discriminant, between both total and factor PCL-5 scores, and B, C, D, E criterions and LEC-5, as well as with criterion A, specifically created for this study.

• Please check the fair usage of a technical term such as 'discriminate' vs. 'discriminant' validity and 'factors' vs. 'factor' vs. 'factorial' validity.

We checked it: the term 'discriminant' remained, with the term ‘divergent’ added in the brackets, while the term 'factorial' was changed to 'factor' throughout the manuscript. Thank you for pointing us to make these corrections. 

• Please check the in-text citation style.

The citation style has been corrected according to the requirements of the journal 

Best regards 

Eirini Orovou

Replies to Reviewers comments 

Dear Reviewer #1:

• The Introduction section is too long. I think that the detailed description of PTSD touching all criteria are unnecessary and should be focused on those details that are appropriate to the results. The historical presentation of PTSD is absolutely an unnecessary surplus in the MS. PTSD is relatively common entity in perinatal context, but the description of the PTSD symptoms, questionnaires should be more concise.

We shortened the Introduction part. The sentence ’So far there are no studies assessing the psychometric properties of the PCL-5, in its Greek translation, applied on a special group of postpartum women’’ was moved from the end of the third paragraph to the end of the ‘Introduction’. Thank you for your suggestions. 

In the Materials and Methods section:

• 'Medical dossier' is not a good expression. Medical charts are the proper word.

The word has been replaced

In the Ethics sub-section: the second meaning is unnecessary ('After ...')

The second meaning deleted

• If the authors carried out test-retest reliability, then they should present the results. 

Unfortunately, there were no test-retest reliability analyses. 

• Were there any back and forth translation and pilot testing of the questionnaire to test the comprehension?

Yes, we did back and forth translation, but not pilot testing of the questionnaire. 

• Description of LEC-5, DSM-5 and PCL-5 can be shortened.

We did it, with an additional small correction of PCL-5, added from a deleted part of the ‘Introduction’: “A provisional PTSD diagnosis can be made if a score of each item is 2 or above (range 0 to 4), when there is as score of one or more in the categories of criteria B and C and two or more in categories D and E, and by summing the score (range 0-80) for each of the 20 items.”

• In Data Analyses: 'Descriptive statistics ...' sentence is unnecessary.

It is deleted.

• The Discussion is a little short and may complete with PCL-5 Factor analysis results in other populations and the consequence of the factor structure. 

• We extended the Discussion, according to your note.

Dear Reviewer #2: 

• I think it need to mention about the limitations of this study more. For example, the author states that "Reported levels of PTSD among participants were low, since the data were positively skewed" in the results, but the effect of this on the results has not been considered. In addition, the participants were two types by Caesarean Sections received, and the impact of this on results should be stated.

We added new information to the ‘Conclusions’, according to your note. Thank you.

Dear Reviewer #3: 

Major concerns:

1)Please describe the translation process of the evaluation scale in more detail. Please show the back translation version of PCL-5 and consider the difference from the original version of one. Moreover, please consider the Linguistically limitations of the Greek version of PCL-5. Also, if there are any points that you paid particular attention to during the translation process, please describe them. In addition, please add the Greek version of PCL-5 as an attachment file (an image file is also acceptable).

Translation process is additionally described. Thank you for directing us to this. I also attach the Greek version of PCL-5 

2)The strength of this study is the large number of samples. On the other hand, too many samples are statistically more likely to cause type I errors (prone to false statistically significant differences). Please estimate the sample size required for this study from prior probabilities and consider whether the number of samples is appropriate.

‘’Suggested minimums for sample size include from 3 to 20 times the number of variablesand absolute ranges from 100 to over1,000.’’ -Daniel J. Mundfrom, Dale G. Shaw & Tian Lu Ke (2005) Minimum Sample Size Recommendations for Conducting Factor Analyses, International Journal of Testing, 5:2, 159-168, DOI: 10.1207/s15327574ijt0502_4. This suggests that our sample size of 469 participants for performing factor analyses is excellent. 

Minor concern:

1) Please describe a brief review of the psychosocial characteristics of the Greek postpartum women after Cesarean Section, comparing them to other countries.

So far in Greece the only research that has been carry out on postpartum disorders in women after cesarean section is our survey, published on March 2020 in MDPI: “Orovou E, Dagla M, Iatrakis G, Lykeridou A, Tzavara C, Antoniou E. Correlation between Kind of Cesarean Section and Posttraumatic Stress Disorder in Greek Women. Int J Environ Res Public Health [Internet]. 2020 Jan [cited 2020 Mar 6];17(5):1592. Available from: https://www.mdpi.com/1660-4601/17/5/1592”

Kind Regards 

Eirini Orovou

---

## [Editor Report · Decision Letter 1]

16 Jun 2021

PONE-D-21-05505R1

Psychometric Properties of the Post Traumatic Stress Disorder Checklist for DSM-5 (PCL-5) in Greek Women after Cesarean Section

PLOS ONE

Dear Dr. Orovou,

Thank you for submitting your manuscript to PLOS ONE. After careful consideration, we feel that it has merit but does not fully meet PLOS ONE’s publication criteria as it currently stands. Therefore, we invite you to submit a revised version of the manuscript that addresses the points raised during the review process.

ACADEMIC EDITOR: Thank you for the revised manuscript that effectively addresses previous comments. However, a few minor edits remain (see my comments below) before this manuscript is considered for publication.

We look forward to receiving your revised manuscript.

Kind regards,

Kenta Matsumura

Academic Editor

PLOS ONE

Journal Requirements:

Additional Editor Comments: Letters and lines are still overlapping in Fig 1 (especially for between "e"s). Please resolve this problem.

---

## [Author Response · Author response to Decision Letter 1]

2 Jul 2021

Dear Editor

Thank you for your comments and suggestions on our manuscript. The changes you suggested are noted below. 

• In the Introduction section the citation [29], which was, “Ford ES. The metabolic syndrome and mortality from cardiovascular disease and all-causes: findings from the National Health and Nutrition Examination Survey II Mortality Study. Atherosclerosis. 2004 Apr;173(2):309–14”, 

was replaced with the right one “Ford E, Ayers S, Bradley R. Exploration of a cognitive model to predict post-traumatic stress symptoms following childbirth. J Anxiety Disord. 2010;24(3):353–9”. The error occurred through the zotero bibliographic system. 

• In the 4th paragraph of the introduction, the citation [19] was omitted as unnecessary

• The reference list is complete and correct. 

• The problem of figure 1 is solved

Kind regards 

Eirini Orovou

---

## [Editor Report · Decision Letter 2]

22 Jul 2021

Psychometric Properties of the Post Traumatic Stress Disorder Checklist for DSM-5 (PCL-5) in Greek Women after Cesarean Section

PONE-D-21-05505R2

Dear Dr. Orovou,

We’re pleased to inform you that your manuscript has been judged scientifically suitable for publication and will be formally accepted for publication once it meets all outstanding technical requirements.

Kind regards,

Kenta Matsumura

Academic Editor

PLOS ONE

Additional Editor Comments:

The authors should provide additional information in their methods section about the participants' demographic characteristics.

"...written consent for their participation in the survey (*N*=.469)."  "... (*N* = 469)"
---

## [Editor Report · Acceptance letter]

28 Jul 2021

PONE-D-21-05505R2 

Psychometric Properties of the Post Traumatic Stress Disorder Checklist for DSM-5 (PCL-5) in Greek Women after Cesarean Section 

Dear Dr. Orovou:

I'm pleased to inform you that your manuscript has been deemed suitable for publication in PLOS ONE. Congratulations! Your manuscript is now with our production department. 

Kind regards, 

on behalf of

Dr. Kenta Matsumura 

Academic Editor

PLOS ONE